# Crustacean Mab21 proteins drive tissue-specific antiviral immunity by activating IKKε outside the canonical nucleic-acid sensing paradigm

Haoyang Li[1,2,3], Qinyao Li[1,2,3], Hao Yang[1,2,3], Xiaodi Wang[1,2,3], Airong Lv[1,2,3], Xuanzheng Di[1,2,3], Ranran Wang[1,2,3], Sheng Wang[1,2,3], Bin Yin[1,2,3], Jianguo He[1,2,3]*, Chaozheng Li ![ORCID][1,2,3]*

1 State Key Laboratory of Biocontrol/ Southern Marine Science and Engineering Guangdong Laboratory (Zhuhai), School of Life Sciences, School of Marine Sciences, Sun Yat-sen University, Guangzhou, China, 2 China-ASEAN Belt and Road Joint Laboratory on Mariculture Technology, Sun Yat-sen University, Guangzhou, China, 3 Guangdong Provincial Key Laboratory of Aquatic Economic Animals, Sun Yat-sen University, Guangzhou, China

* lsshjg@mail.sysu.edu.cn (JH); lichaozh@mail2.sysu.edu.cn (CL)

## Abstract

The Mab21/cGAS protein family has diversified across metazoans to regulate development and innate immunity. In vertebrates, cGAS detects cytosolic DNA and synthesizes 2′3′-cGAMP to activate STING–TBK1–IRF signaling, while invertebrate cGAS-like receptors (cGLRs) recognize RNA or DNA and generate non-canonical cyclic dinucleotides. However, whether shrimp Mab21 proteins function as canonical nucleic acid sensors remains unresolved. Here, we identified three Mab21 proteins from *Litopenaeus vannamei*—LvMab21–1, LvMab21–2, and LvMab21–3. Although they are phylogenetically related to cGAS-like proteins, none bound dsDNA or dsRNA or synthesized cGAMP in response to ISD or poly(I:C). Instead, all three interacted directly with the TBK1 homolog LvIKKε, promoted its phosphorylation at serine 175, and thereby activated the downstream IRF–Vago4 signaling axis. This mechanism defines a non-canonical nucleic acid sensing paradigm, whereby Mab21 proteins act as protein-based enhancers of kinase activation rather than as nucleic acid–dependent CDN synthases. We further show that these proteins display tissue-specific antiviral functions: all three act in hemocytes, LvMab21–1 predominates in hepatopancreas, LvMab21–2 and LvMab21–3 are most critical in gills, and LvMab21–1 and LvMab21–3 cooperate in intestine. Silencing any Mab21 paralog reduced survival and increased white spot syndrome virus (WSSV) burden, underscoring their physiological relevance. Together, our findings expand the known repertoire of innate immune strategies within the Mab21 family, highlight a previously unrecognized non-canonical mechanism of interferon-like activation, and reveal tissue-specific specialization that tailors antiviral responses across shrimp organs. These insights provide both evolutionary context and candidate targets for breeding disease-resistant shrimp.

**Data availability statement:** All relevant data are within the paper and its Supporting information files.

**Funding:** This research is supported by National Key Research and Development Program of China (2022YFF1000304 to JH), National Natural Science Foundation of China (32373158 and 32441085 to CL), Guangdong Basic and Applied Basic Research Foundation (2025A1515012027 to SW) and Science and Technology Planning Project of Guangzhou City (2025A04J5093 to HL). The funders have no role in study design, data collection and analysis, decision to publish, or preparation of the manuscript.

**Competing interests:** The authors have declared that no competing interests exist.

## Author summary

Shrimp aquaculture is threatened by viral diseases such as white spot syndrome virus (WSSV), yet the mechanisms of shrimp antiviral immunity remain poorly understood. In mammals and insects, members of the Mab21/cGAS protein family detect viral DNA or RNA and produce small messenger molecules that trigger interferon responses. We investigated whether shrimp possess similar sensors. We identified three Mab21 proteins in the Pacific white shrimp (*Litopenaeus vannamei*) and found that, unlike their mammalian and insect counterparts, they do not recognize DNA or RNA and do not produce canonical messenger molecules. Instead, these proteins interact with the kinase IKKε and promote its activation, which in turn drives the interferon-like pathway and expression of antiviral effectors. Remarkably, different Mab21 proteins act in different tissues—hemocytes, gills, hepatopancreas, and intestine—allowing shrimp to mount tissue-specific antiviral responses. Silencing these proteins increased viral loads and reduced survival during WSSV infection. Our findings uncover a non-canonical immune strategy in crustaceans and highlight Mab21 proteins as potential targets for breeding disease-resistant shrimp.

## 1 Introduction

The Mab21 (male abnormal 21) family belongs to the nucleotidyltransferase (NTase) fold superfamily and was first identified in *Caenorhabditis elegans* as regulators of cell fate [1]. Orthologs such as Mab21L1 and Mab21L2 are conserved across vertebrates and exhibit distinct but overlapping expression patterns during embryogenesis, where they are essential for organogenesis and tissue differentiation [2–4]. These developmental roles highlight the evolutionary conservation of Mab21 proteins, yet they also raise questions about how this family has diversified across lineages to acquire additional biological functions.

Beyond development, several Mab21 proteins have been co-opted as innate immune sensors. Human cGAS (MB21D1) recognizes cytosolic DNA and catalyzes synthesis of the cyclic dinucleotide (CDN) 2′3′-cGAMP, which activates the stimulator of interferon genes (STING) pathway to induce type I interferons [5]. In invertebrates, *Drosophila* cGLRs respond to viral RNA by producing 3′2′-cGAMP, oyster cGLRs sense dsDNA to generate hybrid 2′3′-cUA, and cnidarian cGLRs synthesize lineage-specific CDNs that activate NF-κB or IRF signaling [6,7]. Despite their differences in ligand specificity and signaling outputs, these proteins share the ability to couple nucleic acid detection with STING-dependent immune activation [5–7]. This evolutionary diversity illustrates the plasticity of the Mab21/cGAS family as immune sentinels.

Crustaceans possess interferon-like signaling systems that are functionally parallel to vertebrate IFN pathways. In *Litopenaeus vannamei* (Pacific white shrimp), viral DNA activates STING, which recruits IKKε and IRF, leading to IRF dimerization,

nuclear translocation, and induction of interferon-like cytokine Vago proteins that restrict white spot syndrome virus (WSSV) and Decapod iridescent virus 1 (DIV1) replication [8]. In *Marsupenaeus japonicus*, Vago proteins act via integrin β3 and FAK-mediated JAK/STAT activation to induce antiviral effectors [9]. These findings establish that the STING–IRF–Vago–JAK/STAT axis functions as a shrimp antiviral system [10]. Although a shrimp Mab21 protein (LvMab21cp) was reported to enhance STING-dependent responses [11], whether Mab21 proteins act as canonical nucleic acid sensors or instead function through alternative mechanisms remains unresolved.

Given the evolutionary link between shrimp Mab21 proteins and cGAS-like receptors, we hypothesized that they might function as nucleic acid sensors or, alternatively, as modulators of interferon-like signaling. To address this, we examined the identity, evolutionary placement, and antiviral roles of Mab21 proteins in *L. vannamei*. Herin, we identified three Mab21 proteins, designated LvMab21–1, LvMab21–2, and LvMab21–3. Unlike vertebrate cGAS or insect cGLRs, these proteins neither sense nucleic acids nor synthesize cGAMP. Instead, they enhance phosphorylation of the kinase LvIKKε, thereby activating the downstream IRF–Vago4 signaling axis. Their contributions are tissue-dependent: all three act in hemocytes, LvMab21–1 predominates in hepatopancreas, LvMab21–2 and LvMab21–3 are most critical in gills, and LvMab21–1 and LvMab21–3 cooperate in intestine. Silencing any of these genes reduced shrimp survival and increased viral loads during WSSV infection. Together, our findings uncover a previously unrecognized mechanism of interferon-like pathway regulation in crustaceans. This discovery expands the functional repertoire of the Mab21 family, highlights evolutionary flexibility in innate immune pathways, and provides molecular targets for antiviral breeding strategies in aquaculture.

## 2 Results

### 2.1 Identification and sequence analysis of LvMab21–1, LvMab21–2, and LvMab21–3

Using the Mab21 domain sequence from human cGAS as a query, we performed BLAST searches against the *L. vannamei* genome (GCA_042767895.1) and identified three sequences encoding proteins containing Mab21 domains, which we designated LvMab21–1, LvMab21–2, and LvMab21–3. LvMab21–1 comprises 1,042 amino acids (predicted molecular mass 119.6 kDa), LvMab21–2 comprises 452 amino acids (49.8 kDa), and LvMab21–3 comprises 374 amino acids (42.5 kDa). The Mab21 domains span amino acid residues 373–716 in LvMab21–1, 84–440 in LvMab21–2, and 68–362 in LvMab21–3. Multiple sequence alignment revealed that the Mab21 domain is highly conserved among these cGAS homologs. However, the positively charged amino acid residues involved in nucleic acid binding are not fully conserved in non-human cGAS homologs. Furthermore, non-human cGAS proteins lack the nucleotidyltransferase-associated zinc-ribbon domain (Fig 1A).

Phylogenetic analysis of cGAS and cGAS-like homologs from vertebrates, arthropods, cnidarians, and mollusks revealed two major clades (Fig 1B). Group 1, represented by the dsDNA sensor HscGAS, and Group 2, represented by the dsRNA sensors DmcGLR1 and DmcGLR2, diverged prior to the emergence of cnidarians. Notably, LvMab21–1, LvMab21–2, and LvMab21–3 clustered within the HscGAS clade (Group 1), distinct from the *Drosophila* cGLR branch (Group 2). This topology suggests that Mab21 family members likely arose before cnidarian divergence and subsequently underwent functional diversification driven by evolutionary pressures.

To further examine structural features, we generated homology models of the Mab21 domains of LvMab21–1, LvMab21–2, and LvMab21–3 using the crystal structure of human Mab21L1 bound to CTP (PDB: 5EOM) as a template. Global model quality scores (QMEANDisCo) were $0.53 \pm 0.05$, $0.88 \pm 0.05$, and $0.80 \pm 0.05$, respectively. Although these proteins lack canonical residues required for ligand recognition, all three models displayed a conserved surface groove with charged surface that may facilitate ligand recognition or protein–protein interactions (Fig 1C).

### 2.2 LvMab21–1, LvMab21–2, and LvMab21–3 do not mediate nucleic acid–induced cGAMP synthesis

Previous studies have shown that dsDNA-activated HscGAS and dsRNA-activated DmcGLR1 generate cGAMP to activate the type I IFN pathway in human cells [5,6]. To test whether shrimp Mab21 proteins function similarly, we expressed

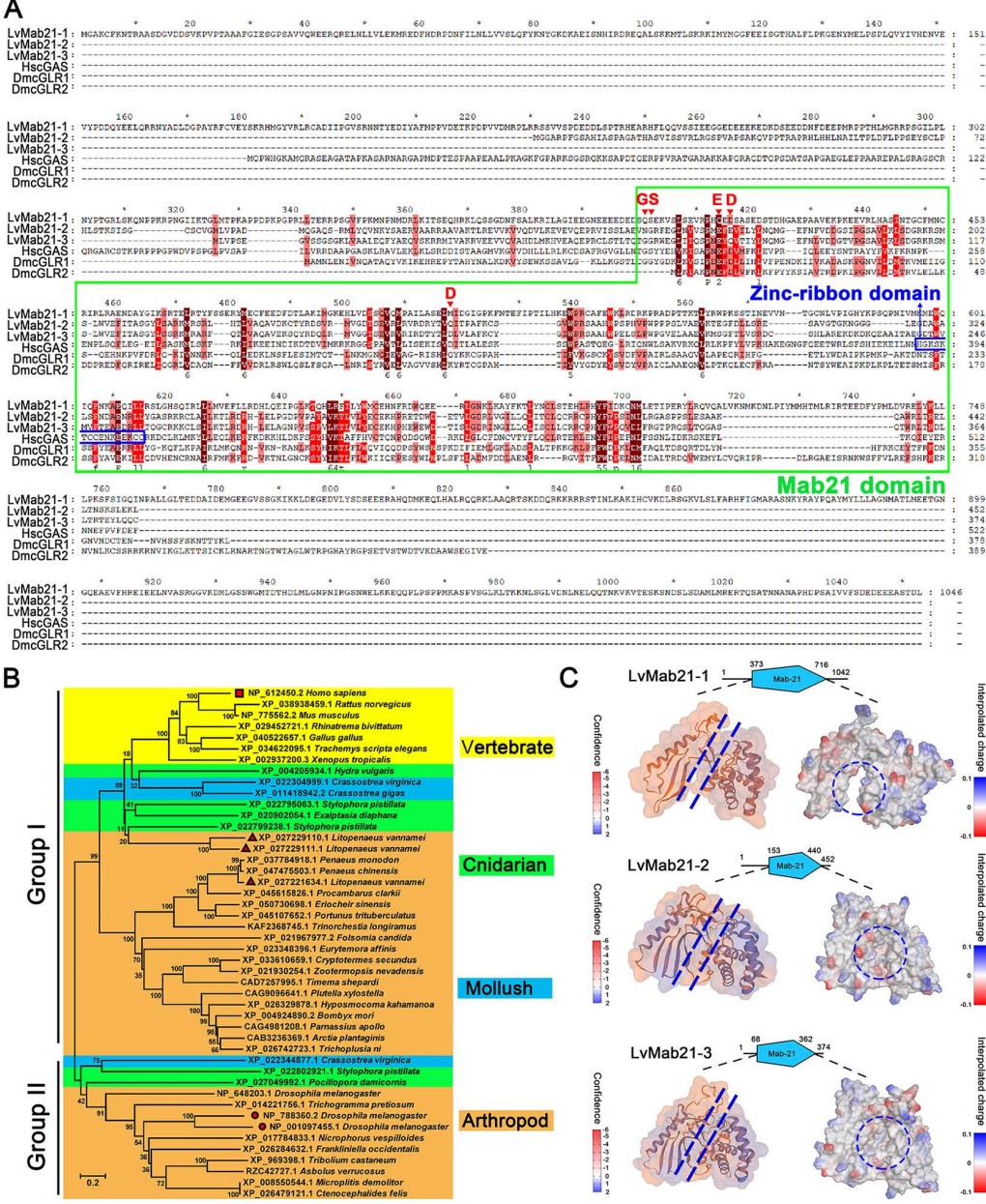

**Fig 1. Sequence alignment, phylogenetic relationships, and structural modeling of Mab21/cGAS homologs.** (A) Multiple sequence alignment of Mab21/cGAS proteins, including LvMab21-1, LvMab21-2, LvMab21-3, HscGAS, DmcGLR1, and DmcGLR2. The conserved Mab21 domains are boxed in solid green lines. The positive positively charged amino acid residues involved in nucleic acid binding are marked via red arrows. The nucleotidyltransferase-associated zinc-ribbon domain are boxed in solid blue lines. (B) Phylogenetic tree constructed from full-length amino acid sequences of cGAS proteins from diverse invertebrate species. HscGAS is marked with a red square; LvMab21-1, LvMab21-2, and LvMab21-3 with red triangles; and DmcGLR1 and DmcGLR2 with red circles. (C) Homology models of Mab21 domains from LvMab21-1, LvMab21-2, and LvMab21-3 generated with SWISS-MODEL and 3D electrostatic potential surface maps predicted by Discovery Studio. Dashed blue lines and dashed blue circles highlight a conserved groove that may serve as a binding site for ligands or interacting proteins.

LvMab21–1, LvMab21–2, or LvMab21–3 together with human STING (HsSTING) in HEK293T-MAVS-KO cells, which lack MAVS and were reconstituted with HsSTING (Figs 2A and S1). HscGAS and DmcGLR1 served as positive controls. Following ISD (dsDNA mimic) stimulation, none of the shrimp Mab21 proteins activated HsSTING signaling to the level observed with HscGAS (Fig 2B and 2C). Likewise, after poly(I:C) (dsRNA mimic) stimulation, LvMab21–1, LvMab21–2,

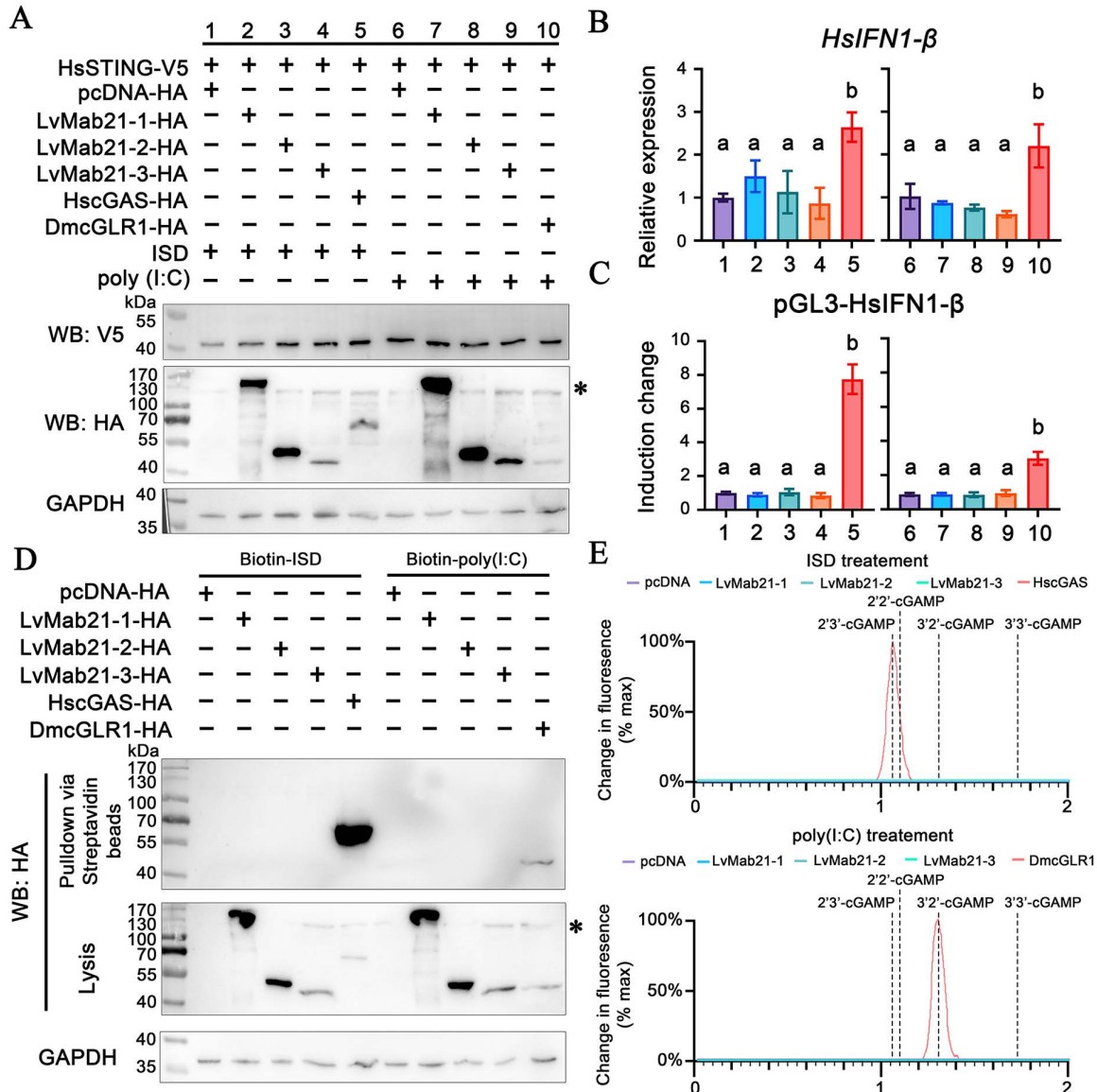

**Fig 2. Shrimp Mab21 proteins do not mediate dsDNA/dsRNA-induced cGAMP synthesis.** (A–C) Effects of shrimp Mab21 protein expression on type I interferon responses. MAVS-knockout HEK293T cells were co-expressed with human STING (HsSTING) and cGAS homologs from shrimp, human, or fruit fly, and stimulated with ISD (dsDNA analog) or poly(I:C) (dsRNA analog). Responses were assessed by immunoblotting (A), quantitative RT-PCR for IFN-β transcripts (B), and IFN-β promoter activity measured by dual-luciferase reporter assay (C). GAPDH served as a loading control. Bars represent mean ± SD ($n = 3$). Different characters indicate significant differences, analyzed by one-way ANOVA. All experiments were repeated independently three times with similar results. (D) Binding of shrimp Mab21 proteins to biotin-labeled ISD or poly(I:C) determined by pull-down assays. (E) Detection of cGAMP production by shrimp Mab21 proteins after ISD or poly(I:C) stimulation, analyzed by mass spectrometry. Asterisks (*) denote non-specific bands.

and LvMab21–3 maintained basal reporter activity and HsIFN1-β transcription, whereas DmcGLR1 robustly activated the HsSTING–IFN pathway (Fig 2B and 2C).

To assess nucleic acid binding, we performed streptavidin bead pull-down assays using biotin-labeled dsDNA or dsRNA. None of the shrimp Mab21 proteins bound to either ligand (Fig 2D). Finally, nucleotide extracts from cells expressing LvMab21 proteins were analyzed by mass spectrometry. Unlike HscGAS (2′3′-cGAMP) and DmcGLR1 (3′2′-cGAMP), no cGAMP was detected in cells expressing LvMab21–1, LvMab21–2, or LvMab21–3 upon ISD or poly(I:C) stimulation (Fig 2E).

## 2.3 LvMab21–1, LvMab21–2, and LvMab21–3 activate LvVago4 through interaction with LvIKKε

Because shrimp Mab21 proteins did not sense nucleic acids or synthesize cGAMP, and given the predicted groove in their Mab21 domains, we hypothesized that they may signal via protein–protein interactions with downstream modulators. Co-immunoprecipitation assays in HEK293T cells revealed that all three Mab21 proteins interacted with LvIKKε (the shrimp homolog of TBK1), but not with LvSTING (Fig 3A). To evaluate functional consequences, dual-luciferase reporter assays were performed by co-expressing LvSTING, LvIRF, and LvIKKε together with individual Mab21 proteins. Compared with empty vector controls, LvMab21–1 enhanced LvVago4 promoter activity ~2.1-fold, LvMab21–2 ~ 1.9-fold, and LvMab21–3 ~ 2.3-fold (Fig 3B). These results indicate that shrimp Mab21 proteins potentiate interferon-like signaling through direct interaction with LvIKKε.

## 2.4 LvMab21–1, LvMab21–2, and LvMab21–3 promote LvIKKε phosphorylation at serine 175

To investigate whether Mab21 proteins influence LvIKKε activation, we first identified phosphorylation sites of LvIKKε by expressing GFP-tagged LvIKKε in HepG2 cells and stimulating with poly(I:C). GFP pull-down followed by LC-MS/MS revealed serine 175 (S175) as the most likely activation site (S2A–S2C Fig; S2, S3 Tables). Sequence alignment confirmed that S175 corresponds to S172, the activating phosphorylation site of human IKKε.

To validate functional importance, we generated an S175A mutant. Dual-luciferase assays showed that wild-type LvIKKε induced ~5.4-fold LvVago4 activation, whereas S175A reduced activity to ~40% of wild-type (Fig 4A). Phospho-specific antibodies recognizing pS175 confirmed phosphorylation in wild-type but not mutant proteins (Fig 4B). Time-course analysis further revealed that LvIKKε phosphorylation at S175 increased in hemocytes during WSSV infection (Fig 4C).

Finally, we tested whether Mab21 proteins affect LvIKKε phosphorylation. Co-expression of LvMab21–1, LvMab21–2, or LvMab21–3 significantly promoted phosphorylation of LvIKKε at S175 compared with vector control (Fig 4D). Thus, Mab21 proteins enhance LvIKKε activation through direct interaction.

## 2.5 LvMab21–1, LvMab21–2, and LvMab21–3 activate the IKKε–IRF–Vago4 pathway in hemocytes

Quantitative RT-PCR revealed distinct tissue-specific expression patterns of LvMab21 genes. LvMab21–1 was most abundant in eyestalk, muscle, hepatopancreas, intestine, and antennal gland, with moderate levels in nerve, heart, and stomach, and low expression in hemocytes and epidermis. LvMab21–2 was highest in muscle and moderately expressed in epidermis, heart, nerve, intestine, and gill, with lower levels in eyestalk, stomach, hemocytes, and antennal gland. LvMab21–3 was strongly expressed in eyestalk and moderately in muscle, antennal gland, and nerve, but weakly expressed in stomach, heart, intestine, hemocytes, and gill (Fig 5A).

Given the role of hemocytes as macrophage-like immune cells, we examined Mab21 responses in this tissue. Upon WSSV infection or poly(I:C) stimulation, LvMab21–1 showed rapid and strong induction (peaking ~10-fold at 36 h for WSSV and ~11-fold at 4 h for poly(I:C)). LvMab21–2 displayed delayed upregulation (peaking ~4–5-fold at 48 h), whereas LvMab21–3 responded rapidly but more modestly (peaking ~3-fold at 8 h for WSSV and ~2-fold at 4 h for poly(I:C)). PBS-treated controls remained at baseline (Fig 5B and 5C).

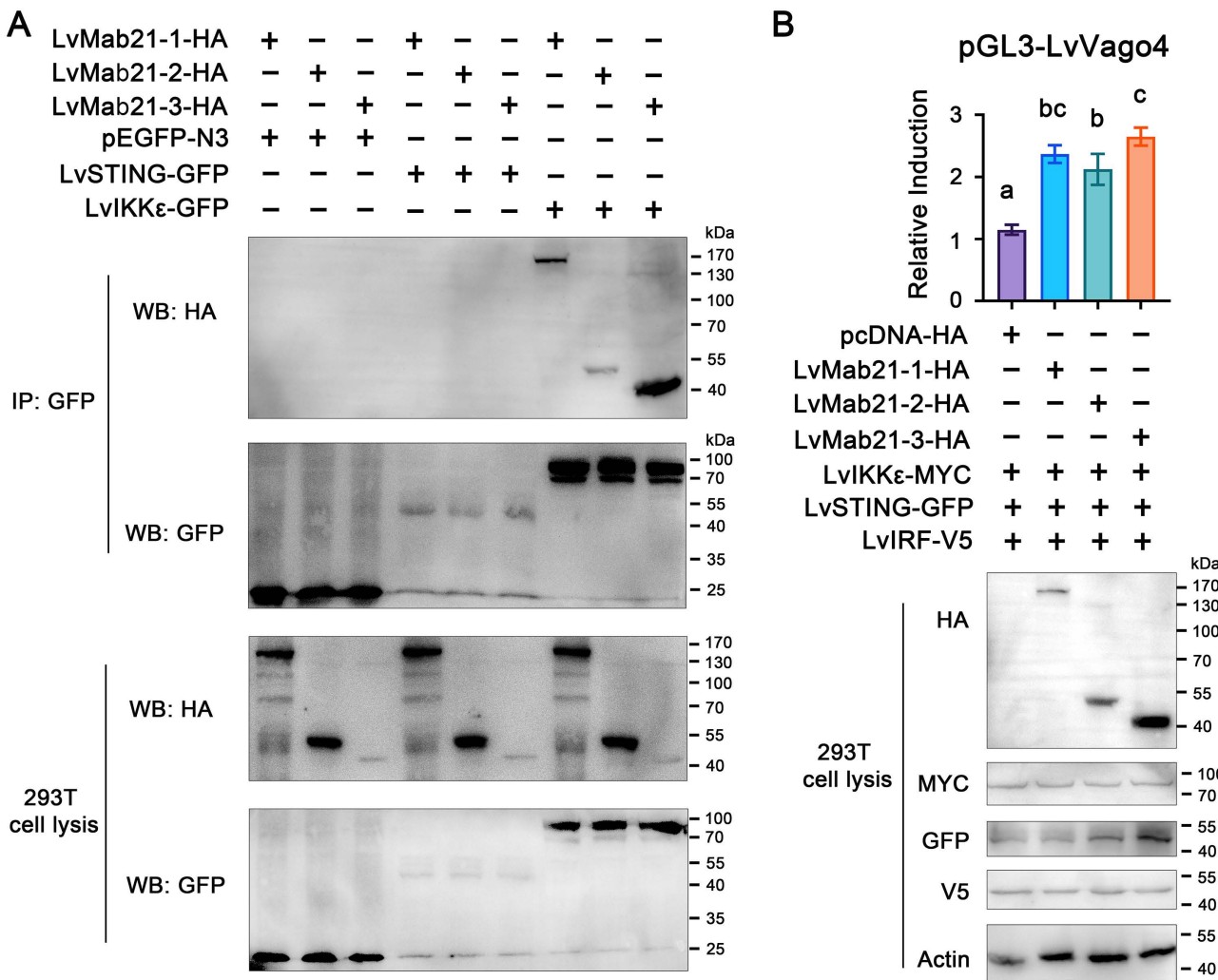

**Fig 3. Shrimp Mab21 proteins enhance the STING–IKKε–IRF–LvVago4 signaling axis in HEK293T cells.** (A) Interactions between GFP-tagged LvSTING or LvIKKε and HA-tagged LvMab21-1, LvMab21-2, or LvMab21-3 were examined by co-immunoprecipitation (Co-IP) assays. (B) Relative induction of LvVago4 promoter activity mediated by LvSTING-GFP, LvIKKε-MYC, LvIRF-V5, and HA-tagged LvMab21-1, LvMab21-2, or LvMab21-3. Protein expression was confirmed by Western blotting, and promoter activity was measured by dual-luciferase reporter assays. Actin served as a loading control. Bars represent mean ± SD ($n = 3$). Different characters indicate significant differences, analyzed by one-way ANOVA. All experiments were independently repeated three times with similar results.

Functional assays using RNAi confirmed the importance of Mab21 proteins in hemocyte antiviral responses. Knockdown of LvMab21–1, LvMab21–2, or LvMab21–3 significantly reduced their transcript levels after WSSV infection (Fig 6A and 6B). Each knockdown also suppressed LvIKKε phosphorylation at S175 (Fig 6C), impaired WSSV-induced LvIRF dimerization and nuclear translocation (Fig 6C and 6D), and reduced LvVago4 expression (Fig 6B). Collectively, these results demonstrate that all three shrimp Mab21 proteins positively regulate the IKKε–IRF–Vago4 pathway in hemocytes.

## 2.6 Tissue-specific regulation of the IKKε–IRF–Vago4 pathway by LvMab21–1, LvMab21–2, and LvMab21–3

Because the gill serves as the primary entry site for WSSV and both hepatopancreas and intestine express high levels of LvSTING and LvIRF, we examined Mab21 function in these tissues. RT-qPCR revealed distinct expression dynamics

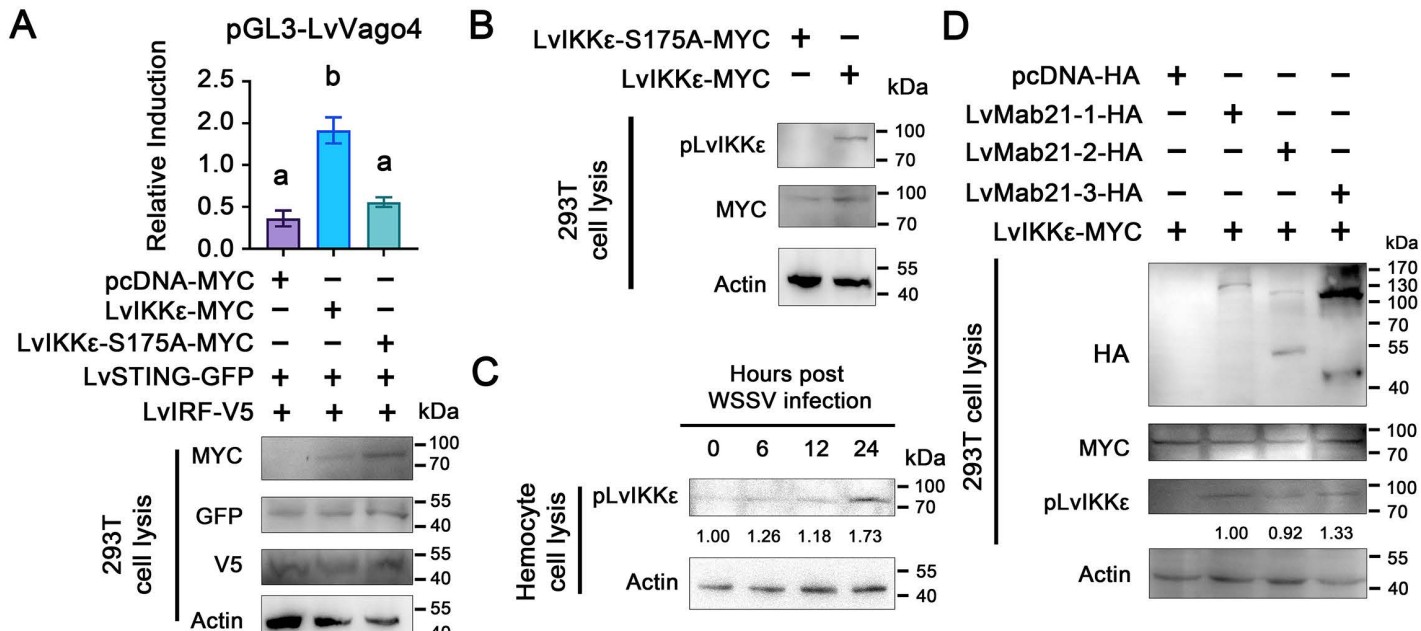

**Fig 4. Shrimp Mab21 proteins promote LvIKKε phosphorylation at serine 175.** (A)Relative induction of LvVago4 promoter activity mediated by LvIRF-V5 and either wild-type LvIKKε-MYC or mutant LvIKKε-S175A-MYC in HEK293T cells. Proteins were ectopically expressed and verified by Western blotting, and promoter activity was measured by dual-luciferase assays. Actin served as a loading control. Bars represent mean±SD ($n$=3). Different characters indicate significant differences, analyzed by one-way ANOVA. (B) Validation of a phospho-specific antibody recognizing LvIKKε phosphorylated at serine 175 (pS175). HEK293T cells expressing wild-type (LvIKKε-MYC) or mutant (LvIKKε-S175A-MYC) proteins were probed with anti-pS175 antibody. (C) Time-course detection of LvIKKε phosphorylation in shrimp hemocytes during WSSV infection. (D) Detection of LvIKKε phosphorylation in HEK293T cells ectopically expressing LvMab21-1-HA, LvMab21-2-HA, or LvMab21-3-HA. The blotting band intensity was also digitated using ImageJ software (C and D). The relative protein level was shown as the ratio between pho-LvIKKε and actin. All experiments were independently repeated three times with similar results.

following immune stimulation. In hepatopancreas infected with WSSV, LvMab21−1 was rapidly induced (peaking at 8 h) but declined below baseline by 72 h, whereas LvMab21−2 and LvMab21−3 remained low until significantly upregulated at 48 h (Fig 7A). In gills, all three Mab21 genes were induced by poly(I:C) at 8 h and then declined to baseline (S3B Fig), while WSSV triggered early induction (4 h) followed by oscillatory fluctuations (Fig 7B). In intestine stimulated by WSSV, LvMab21−1 and LvMab21−2 remained low except for modest induction at 24 h, whereas LvMab21−3 was upregulated as early as 4 h (Fig 7C).

RNAi experiments further revealed tissue-specific regulation of the interferon-like pathway. At 48 h post-WSSV infection, LvVago4 expression in hepatopancreas was most strongly suppressed by LvMab21−1 knockdown (~0.14-fold), with more moderate effects of LvMab21−2 and LvMab21−3 knockdown (~0.56- and ~0.54-fold, respectively). In gills, silencing LvMab21−1, LvMab21−2, or LvMab21−3 reduced LvVago4 expression by 63%, 23%, and 21%, respectively. In intestine, LvMab21−2 knockdown had the strongest effect (62% reduction), whereas LvMab21−1 and LvMab21−3 knockdown reduced expression by 33% and 22%, respectively (Fig 8A). Consistently, knockdown of Mab21 proteins attenuated LvIKKε phosphorylation and LvIRF dimerization in a tissue-dependent manner: LvMab21−1 in hepatopancreas, LvMab21−2 and LvMab21−3 in gills, and LvMab21−1 and LvMab21−3 in intestine (Fig 8B).

The antiviral relevance of Mab21 proteins was further confirmed through in vivo challenge experiments. Survival analysis revealed that shrimp subjected to RNAi-mediated knockdown of LvMab21−1, LvMab21−2, or LvMab21−3individually showed significantly lower survival rates following WSSV infection compared to

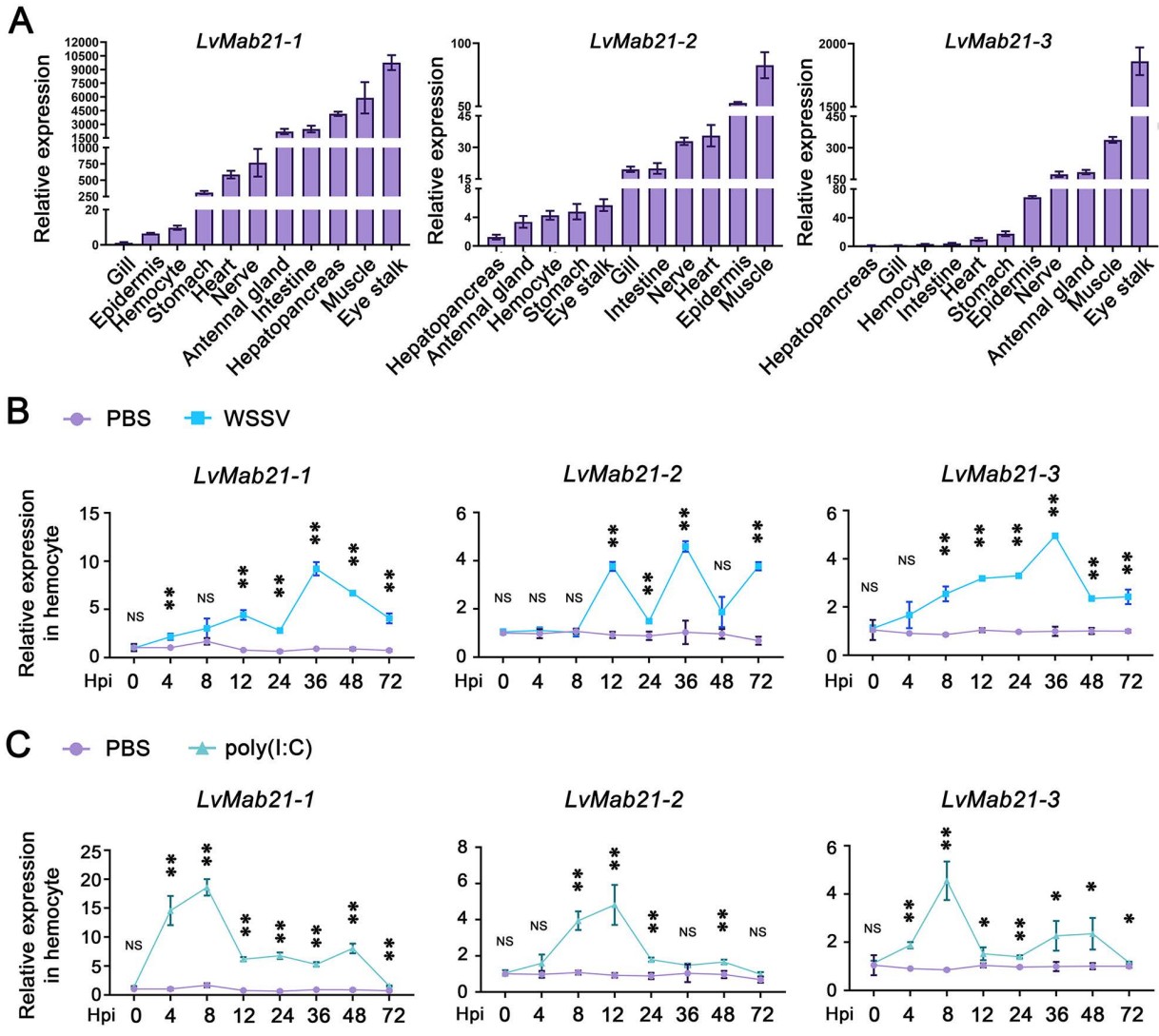

**Fig 5. Expression of LvMab21-1, LvMab21-2, and LvMab21-3 in healthy and immune-stimulated shrimp.** (A) Tissue distribution of LvMab21-1, LvMab21-2, and LvMab21-3 transcripts in *L. vannamei*. EF-1α served as the internal reference gene. Data are presented as mean ± SD from three independent assays. Expression levels in gill (LvMab21-1), hepatopancreas (LvMab21-2), and hepatopancreas (LvMab21-3) were each used as calibrators and set to 1.0. (B-C) Expression profiles of LvMab21-1, LvMab21-2, and LvMab21-3 in hemocytes following PBS, WSSV, or poly(I:C) injection. Expression levels at each time point were normalized to those of the 0 h PBS-injected control group. Data in (B) and (D) are presented as mean ± SD and comparison between the experimental group and the control group at each time point was performed using Student's *t*-test (NS, not significant, *$p < 0.05$, **$p < 0.01$).

dsGFP-injected controls (log-rank test: χ² = 7.312, $p = 0.0069$; χ² = 6.243, $p = 0.0125$; χ² = 6.280, $p = 0.0122$, respectively; Fig 9A). Notably, simultaneous knockdown of all three Mab21 genes resulted in an even more pronounced reduction in survival, with statistical significance also evident when compared to each individual knockdown group (log-rank test: χ² = 8.484, $p = 0.0036$; χ² = 10.58, $p = 0.0011$; χ² = 10.51, $p = 0.0012$, respectively). Consistently, viral DNA loads in gill tissues were markedly elevated upon Mab21 silencing, increasing approximately 3.99-fold, 2.22-fold, 3.60-fold, and 6.88-fold in the LvMab21–1, LvMab21–2, LvMab21–3 single knockdown, and the triple-knockdown groups, respectively (Fig 9B).

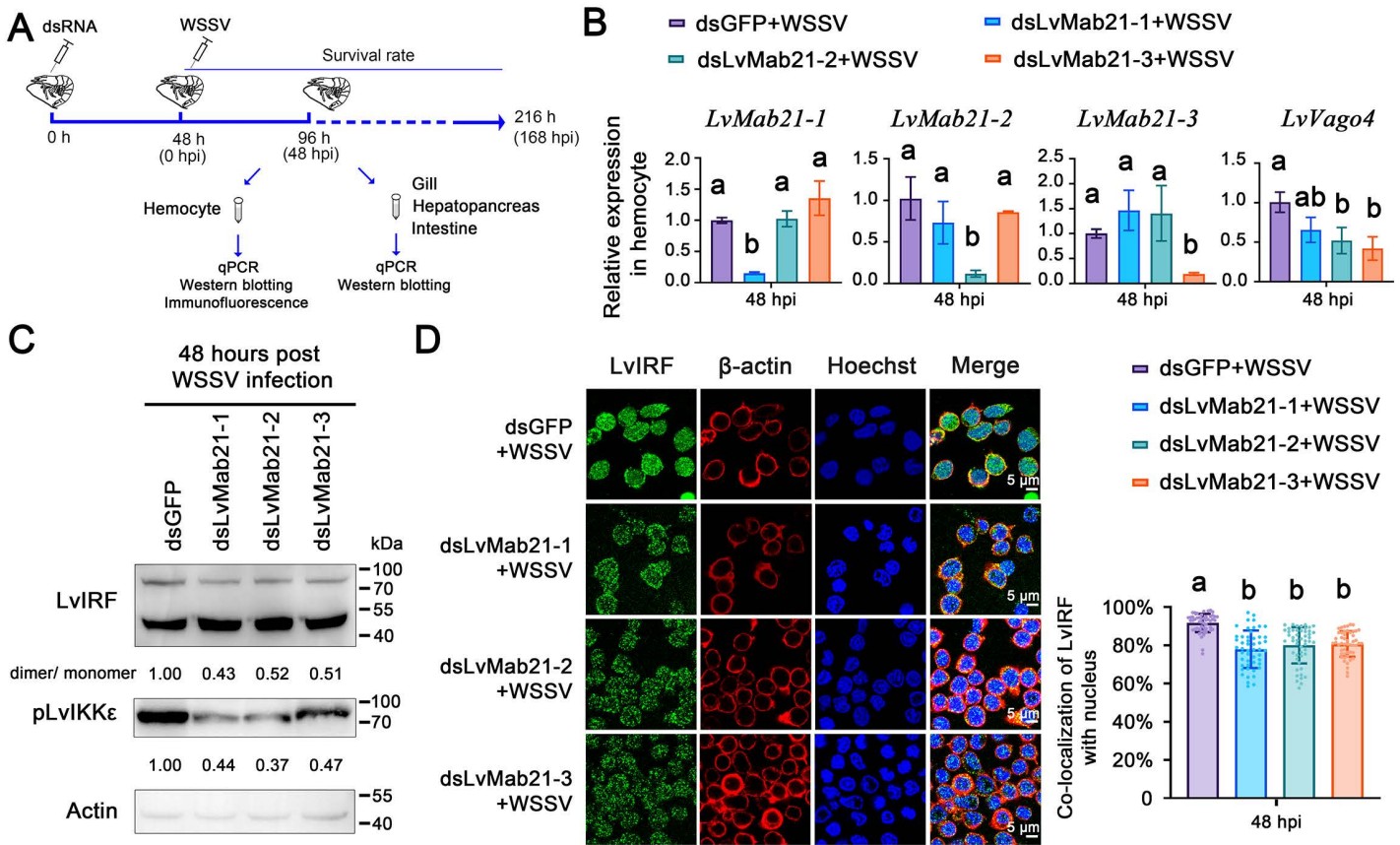

**Fig 6. LvMab21-1, LvMab21-2, and LvMab21-3 activate the IKKε–IRF–Vago4 pathway in hemocytes.** (A) Schematic representation of the procedures used to investigate the potential antiviral functions of LvMab21-1, LvMab21-2, and LvMab21-3 *in vivo*. (B) RNAi knockdown efficiencies and LvVago4 expression levels in hemocytes during WSSV challenge, measured by qRT-PCR. (C) Detection of LvIRF dimerization and LvIKKε phosphorylation in hemocytes by Western blotting. The blotting band intensity was also digitated using ImageJ software. The relative protein level was shown as the ratio between LvIRF dimer and monomer, or pho-LvIKKε and actin. (D) Nuclear translocation of LvIRF observed by immunofluorescence. Co-localization of LvIRF with the nucleus was quantified in hemocytes (*n* = 50 cells). Data in (B) and (D) are presented as mean ± SD and were using ordinary one-way ANOVA with multiple comparisons. All experiments were independently repeated three times with similar results.

Together, these findings demonstrate that shrimp Mab21 proteins regulate the interferon-like pathway in a tissue-specific manner. In hemocytes, all three proteins enhance pathway activation; in hepatopancreas, LvMab21–1 is the major contributor; in gills, LvMab21–2 and LvMab21–3 dominate; and in intestine, LvMab21–1 and LvMab21–3 act cooperatively. This tissue-specific modulation highlights the diverse functional roles of Mab21 proteins in shrimp antiviral immunity.

## 3 Discussion

The Mab21 protein family has diversified considerably across metazoans, with functions ranging from developmental regulation to innate immune sensing. In mammals, human cGAS (MB21D1) recognizes cytosolic DNA and produces 2′3′-cGAMP to activate STING-dependent interferon signaling [12]. In invertebrates, homologous cGAS-like receptors (cGLRs) have adapted to detect different nucleic acid ligands and generate non-canonical cyclic dinucleotides, including 3′2′-cGAMP in Drosophila and hybrid 2′3′-cUA in oysters [6,7]. These discoveries highlight the evolutionary flexibility of the Mab21/cGAS family as immune sentinels. Our study expands this landscape by showing that three Mab21 proteins in *L. vannamei*, LvMab21–1, LvMab21–2, and LvMab21–3, were not observed to sense nucleic acids tested in this study,

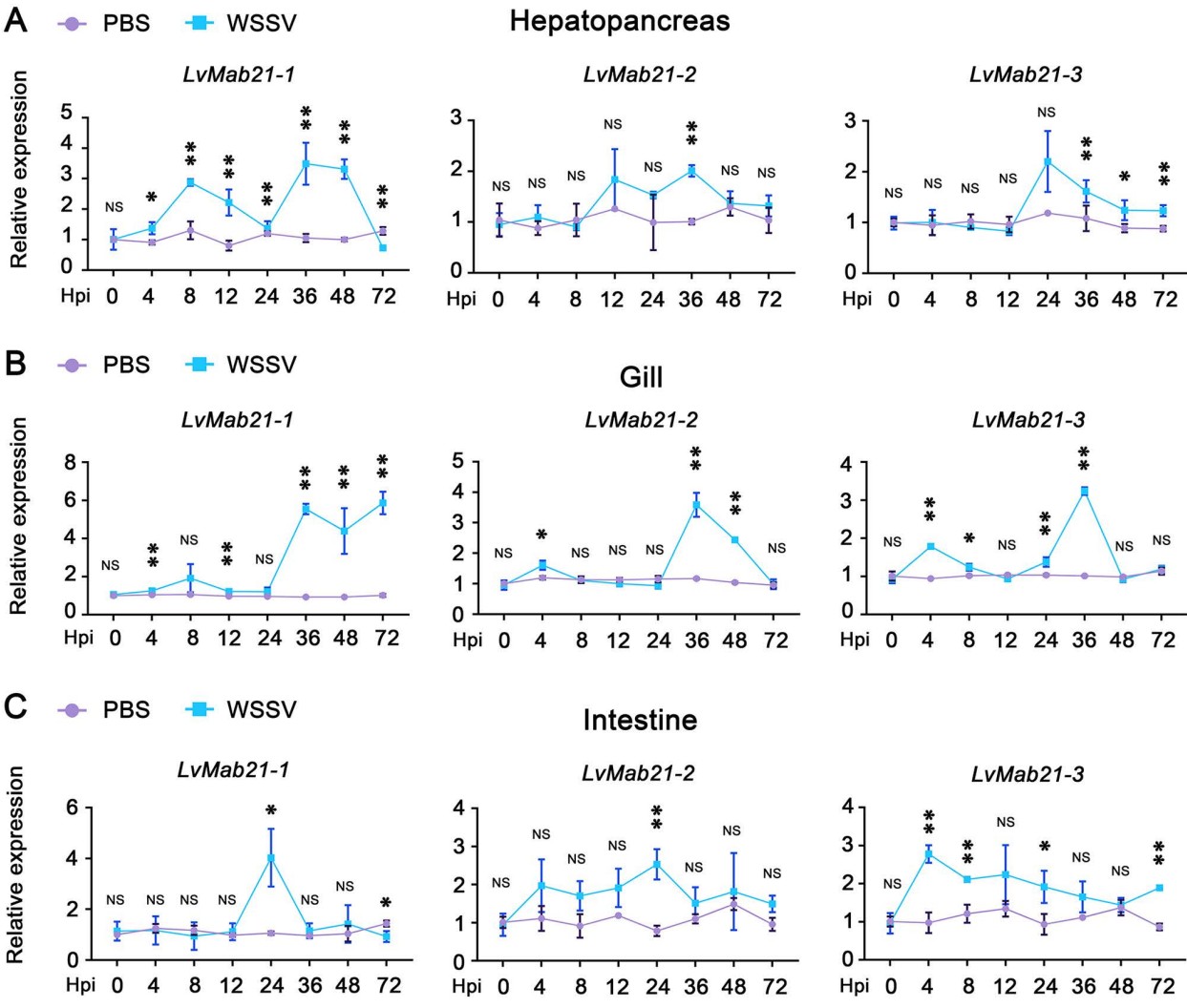

**Fig 7. LvMab21-1, LvMab21-2, and LvMab21-3 respond to WSSV stimulation in hepatopancreas, gill, and intestine.** Expression levels of LvMab21-1, LvMab21-2, and LvMab21-3 in shrimp hepatopancreas (A), gill (B), and intestine (C) following WSSV injection were measured by qRT-PCR. Expression values at each time point were normalized to those of the 0 h PBS-injected control group. Data are presented as mean ± SD and comparison between the experimental group and the control group at each time point was performed using Student's $t$-test (NS, not significant, $*p < 0.05$, $**p < 0.01$).

including ISD and poly(I:C), or synthesize detectable cGAMP under our experimental conditions, but instead appear to activate the interferon-like pathway through direct interaction with LvIKKε. This finding suggests a previously unrecognized, non-canonical mechanism of Mab21-mediated immune regulation in crustaceans, though further studies may reveal additional ligand specificities or signaling outputs.

Although our experiments did not detect nucleic acid binding or canonical cGAMP synthesis by LvMab21–1, LvMab21–2, or LvMab21–3 in response to ISD or poly(I:C), these results do not exclude the possibility that shrimp Mab21 proteins recognize other nucleic acid structures or produce non-canonical cyclic dinucleotides. For example, cnidarian and mollusk cGLRs generate lineage-specific CDNs distinct from 2′3′-cGAMP [7], while *Drosophila* cGLR1 evolved specificity for long dsRNA and produces 3′2′-cGAMP [6]. It is therefore plausible that shrimp Mab21 proteins may act on alternative ligands—such as ssRNA, DNA–RNA hybrids, or structured nucleic acids—or synthesize unconventional signaling molecules that

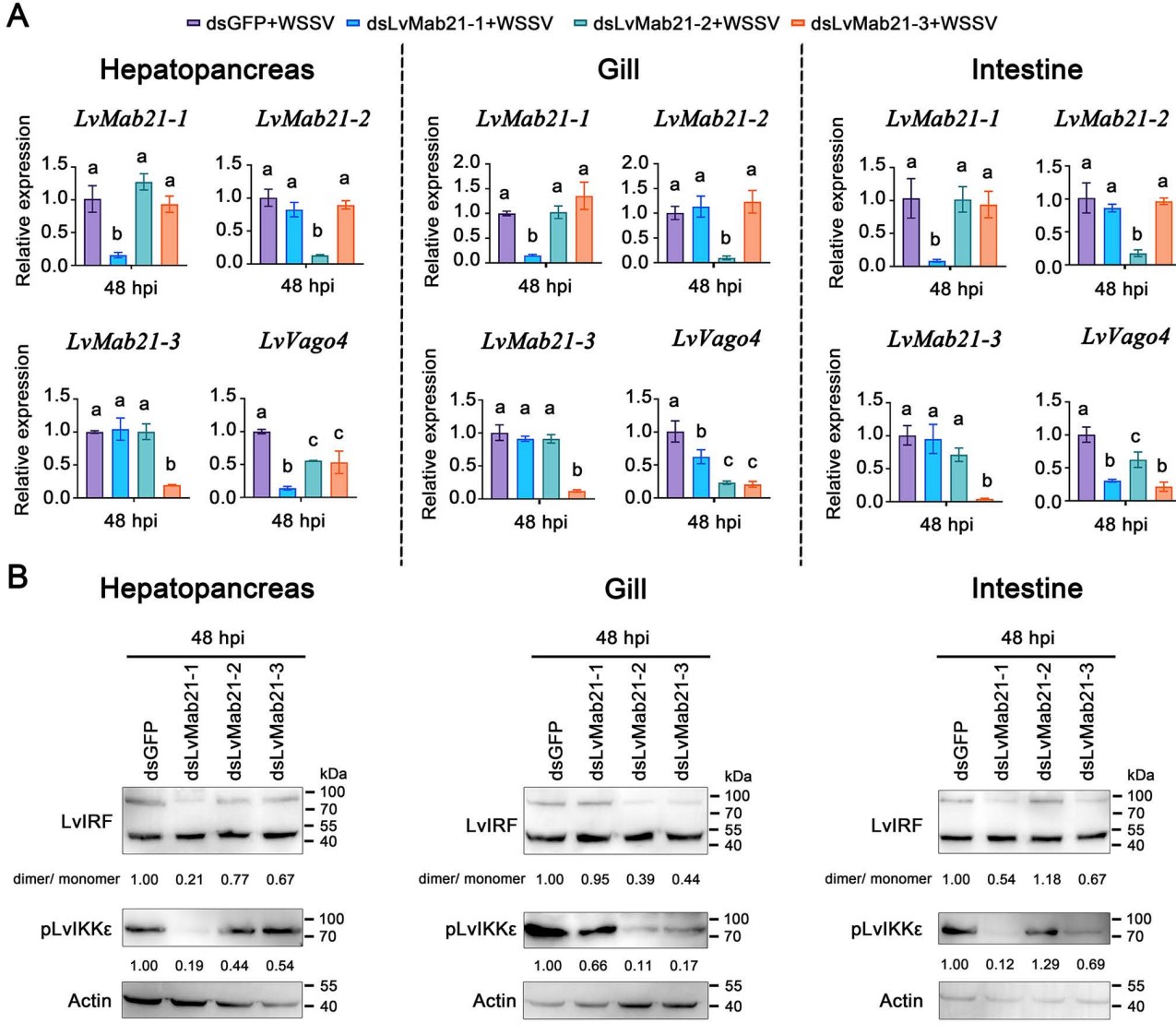

**Fig 8. LvMab21-1, LvMab21-2, and LvMab21-3 activate the STING–IKKε–IRF–Vago4 pathway in hepatopancreas, gill, and intestine in a tissue-specific manner.** (A) RNAi knockdown efficiencies and LvVago4 expression levels in hepatopancreas, gill, and intestine during WSSV challenge, measured by qRT-PCR. Data are presented as mean±SD and were using ordinary one-way ANOVA with multiple comparisons. (B) Detection of LvIRF dimerization and LvIKKε phosphorylation in hepatopancreas, gill, and intestine by Western blotting. The blotting band intensity was also digitated using ImageJ software. The relative protein level was shown as the ratio between LvIRF dimer and monomer, or pho-LvIKKε and actin.

were not captured by our assays. Further biochemical and structural studies will be required to explore these possibilities. Our current findings thus define a clear functional distinction from vertebrate cGAS and insect cGLRs, while leaving open the potential for novel recognition capacities that remain to be discovered.

The mechanistic insight gained here highlights important evolutionary distinctions. Canonical cGAS activation relies on DNA binding via zinc-ribbon or positively charged motifs [13], followed by cyclic dinucleotide synthesis that induces STING oligomerization and TBK1 recruitment [14]. By contrast, shrimp Mab21 proteins lack essential nucleotidyltransferase residues, and our structural analyses revealed no evidence of enzymatic activity. Instead, all three proteins bind LvIKKε and enhance its phosphorylation at serine 175, a site conserved with human IKKε S172. This represents a direct route to IRF

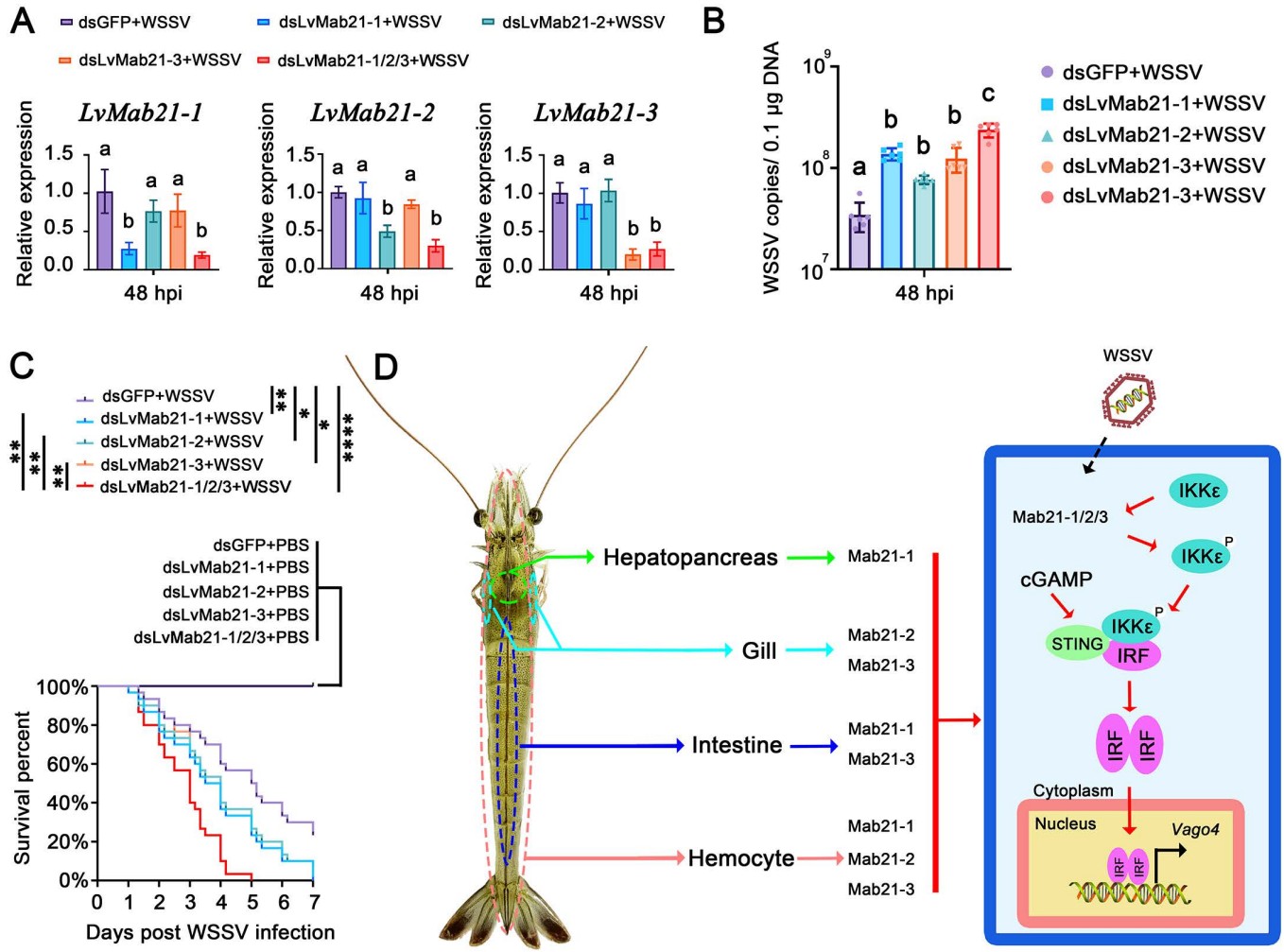

**Fig 9. LvMab21-1, LvMab21-2, and LvMab21-3 are essential for host defense against WSSV infection in shrimp.** (A) RNAi knockdown efficiencies in gill during WSSV challenge, measured by qRT-PCR. Data are presented as mean ± SD and were using ordinary one-way ANOVA with multiple comparisons. (B) WSSV viral loads in gill at 48 h post-infection. Viral genome copies were quantified by qPCR and normalized to 1 µg of total DNA. Data are presented as mean ± SD and were using ordinary one-way ANOVA with multiple comparisons. (C) Survival curves of WSSV-infected shrimp injected with dsRNA. Survival was monitored every 4 h. Statistical significance was determined by log-rank test (*$p < 0.01$; **$p < 0.001$; ****$p < 0.00001$). (D) Proposed model of tissue-specific regulation of antiviral immunity by shrimp Mab21 proteins. Upon WSSV infection, shrimp Mab21 proteins interact with LvIKKε and promote its phosphorylation at serine 175. Activated LvIKKε in turn stimulates downstream IRF activation, leading to IRF dimerization, nuclear translocation, and induction of the interferon-like effector Vago4. Unlike vertebrate cGAS or insect cGLRs, shrimp Mab21 proteins do not function as classical nucleic acid sensors or produce cGAMP, but instead act as direct activators of IKKε. Their contributions are tissue-specific: LvMab21-1 predominates in hepatopancreas, LvMab21-2 and LvMab21-3 are most critical in gill, and LvMab21-1 and LvMab21-3 cooperate in intestine. In hemocytes, all three paralogs contribute to activation of the IKKε–IRF–Vago4 pathway, ensuring broad antiviral protection.

activation and Vago induction, bypassing the requirement for cyclic dinucleotide messengers. Such diversification demonstrates how the Mab21 family has evolved distinct signaling strategies: nucleotide-based messengers in vertebrates and mollusks, versus protein–protein interaction modules in shrimp.

A notable feature of shrimp Mab21 proteins is their tissue-specific functional specialization. Expression and RNAi analyses revealed that all three Mab21s act in hemocytes, LvMab21–1 plays the dominant role in hepatopancreas, LvMab21–2 and LvMab21–3 are critical in gills, and LvMab21–1 and LvMab21–3 cooperate in intestine. This pattern likely reflects the

compartmentalized nature of shrimp antiviral immunity: hemocytes serve as phagocytic immune cells [15], gills are the primary site of pathogen entry [16], and hepatopancreas and intestine are key immune-metabolic tissues [17,18]. Such differential regulation enables shrimp to mount tailored antiviral responses while minimizing immune trade-offs. Importantly, knockdown of any Mab21 paralog reduced survival and increased WSSV loads, demonstrating that their contributions, while tissue-specific, are all physiologically relevant.

These findings also suggest practical applications for shrimp breeding. In aquaculture, antiviral pathways mediated by interferon-like signaling can antagonize antibacterial NF-κB responses [19]. Enhancing pathway activation in a tissue-specific manner may allow selective strengthening of antiviral defenses without compromising antibacterial capacity. Our discovery of Mab21 paralogs as tissue-targeted enhancers of the IKKε–IRF–Vago4 axis therefore provides promising candidate genes for molecular design breeding. Similar to targeted genetic improvements in livestock (e.g., IGF2 editing in pigs to balance lean growth and intramuscular fat) [20] and crops (e.g., tissue-specific complementation in sorghum to decouple growth from digestibility) [21], shrimp Mab21 proteins could be exploited to optimize multi-trait resistance.

In conclusion, this study identifies three shrimp Mab21 proteins as non-canonical regulators of the interferon-like pathway. Instead of functioning as nucleic acid sensors, they act as protein-based activators of IKKε, with tissue-specific patterns of antiviral regulation. These findings expand our understanding of innate immune diversification within the Mab21 family, provide mechanistic insight into shrimp antiviral immunity, and open new avenues for breeding strategies aimed at improving disease resistance. Future work should focus on structural characterization of Mab21–IKKε interactions, comparative studies across crustaceans to define evolutionary conservation, and applied testing of Mab21 variants for resistance breeding in aquaculture.

## 4 Materials and methods

**Ethics statement.** All animal experiments were approved by Institutional Animal Care and Use Committee of Sun Yat-Sen University (SYSU-IACUC-2023-B0005).

### 4.1 Animals, pathogens and immune stimulants

Healthy shrimp (5 ± 0.5 g) were obtained from Guangdong Hisenor Group Co., Ltd Guangzhou, P. R. China and maintained in aerated seawater (5‰ salinity 25 ℃) of a recirculating aquaculture system with a water filtration plant, automatic temperature control equipment, and an ultraviolet sterilizer. Water was totally changed every 12 h. The commercial food (HAID Group) was supplied to the shrimp three times each day, and the leftover feed and excretions were removed in time. The healthy shrimp acclimated for three days before the injection experiments. Before all experiment treatments, shrimp (5% of total) were detected and confirmed to be free of common pathogens including WSSV, DIV1, Yellow head virus, Taura syndrome virus, Infectious hypodermal and hematopoietic necrosis virus and *Vibrio parahaemolyticus* by PCR or RT-PCR methods according to standard operation procedures by Panichareon *et al.* [22] and Qiu *et al.* [23].

Because WSSV was one of the main DNA viruses threatening shrimp culture, they thus were used here as potential activators for the shrimp STING-IKKε-IRF-Vago4 pathway. WSSV particles were extracted from the muscle tissue of freshly WSSV-infected shrimp and stored at −80 °C according to a published method [8]. Before injection, a WSSV inoculum was prepared to approximately $1 \times 10^5$ copies in 50 μL PBS. In the pathogenic challenge experiments, each shrimp received intraperitoneal injection of 50 μL of WSSV inoculum at the second abdominal segment with a 1-mL syringe.

ISD and poly(I:C) were selected as immune stimulants. Each stimulant was diluted to a final concentration of 1 μg/mL in culture medium and transfected into cells using high-efficiency transfection reagent 2.0 (MIKX), followed by incubation for 6 h.

## 4.2 Bioinformatics

To obtain the cDNA sequence of *Mab21–1*, *Mab21–2*, *Mab21–3* from shrimp, the amino acid (aa) sequences of the cGAS homologs from *Human* (Genbank No. NM_138441.3) were collected and used as query sequences for the *in silico* search of *L. vannamei* genome data by using local TBLASTN alignment tool with E-value cutoff of $1e^{-5}$.

Genome and transcript sequences of cGAS or cGAS-like homologs from other species were retrieved from the National Center for Biotechnology Information (NCBI, http://www.ncbi.nlm.nih.gov/). Sequence alignments were analyzed using Clustal X v2.0 [24]. Phylogenetic trees were constructed according to the deduced amino acid sequences with MEGA 5.0 software. Amino acid substitution type and Poisson model and bootstrapping procedure with a minimum of 1000 bootstraps were used [25]. To elucidate the structural characteristics of Mab21 family proteins in *L. vannamei*, the three-dimensional (3D) structures of the conserved Mab21 domains from three putative Mab21 family members (LvMab21–1, LvMab21–2, and LvMab21–3) were predicted *in silico*. Homology modeling was performed using SWISS-MODEL (https://swissmodel.expasy.org) [26]. 3D electrostatic potential surface map was established by Discovery Studio 2021 [27].

## 4.3 Plasmids construction

The coding sequences of LvMab21–1 (XP_027221634.1), LvMab21–2 XP_027229110.1), LvMab21–3 (XP_027229111.1), HsSTING (NP_938023.1), HscGAS (NP_612450.2), DmcGLR1 (NP_788360.2), LvSTING (AQZ41234.1), LvIRF (AKG54423.1) and LvIKKε (AEK86520.1) were cloned into HEK293T cell expressing plasmids, pcDNA3.1A, pcDNA-HA, pcDNA-MYC and pEGFP-N3 vectors, for V5, HA, MYC and GFP-labeled proteins. pcDNA3.1A, pcDNA-HA, pcDNA-MYC, or pEGFP-N3 vectors for expressing V5, HA, MYC, or GFP tagged proteins. The expression plasmids encoding the sequences of the mutant of LvIKKε were constructed through overlap extension PCR and cloned into pcDNA-MYC. The primers used are listed in S1 Table.

## 4.4 Western blotting

HEK293T MAVS-knockout cells were seeded in 6-well plates at a density of $2 \times 10^5$ cells per well in complete growth medium. Transfections were performed 24 hours post-seeding using Superluminal transfection reagent (MIKX) according to the manufacturer's instructions. Specifically, a transfection mixture containing 1 μg V5-tagged plasmid and 1 μg HA-tagged plasmid was prepared by diluting the DNA to a final concentration of 10 ng/μL in serum-containing DMEM medium. The Superluminal reagent was added at a ratio of 3 μL per μg of total plasmid DNA. The solution was vortex-mixed thoroughly and incubated at room temperature for 10 minutes to allow complex formation. The DNA-reagent complexes were then added dropwise to the cells in fresh complete medium. In brief, 48 h after transfection, HEK293T MAVS-knockout cells were harvested and washed with ice-cold PBS three times and then lysed in IP lysis buffer (Pierce) with Halt Protease Inhibitor Cocktail (Thermo fisher scientific). The protein samples were quantified via using BCA assay, mixed with 5 × loading buffer, and denatured by boiling at 100 °C for 5 min. Following SDS-PAGE electrophoresis, proteins were transferred to PVDF via wet transfer, blocked with 5% non-fat milk in TBST, and incubated with the primary antibody, including rabbit anti-V5 antibody (Proteintech), rabbit anti-GFP antibody (Proteintech), rabbit anti-HA antibody (CST), GAPDH Antibody (Proteintech) and mouse β-Actin Antibody (Abmart) for two hours at room temperature. Membranes were then incubated with species-matched HRP-conjugated secondary antibodies including Anti-Rabbit IgG (H + L), HRP Conjugate (Promega) and Anti-Mouse IgG (H + L), HRP Conjugate (Promega) for one hour. Signals were detected using the Omni-ECL kit (Epizyme) and imaged on a Tanon imaging system.

## 4.5 qRT-PCR

The mRNA levels of genes in the HEK293T MAVS-knockout cells expressing targeted proteins, pathogenic challenge experiments or *in vivo* RNA interference (RNAi) experiments were assessed with qRT-PCR assays. The expression levels

of *HsIFN1-β*, *LvMab21–1*, *LvMab21–2*, *LvMab21–3* and *LvVago4* were detected using a LightCycler 480 System (Roche, Basel, Germany) in a final reaction volume of 10 µL, which comprised 1 µL of 1:10 cDNA diluted with ddH$_2$O, 5 µL of GoTaq qPCR Master Mix (Promega), and 250 nM primer (S1 Table). The cycling program was as follows: one cycle of 95 °C for 2 min, followed by 40 cycles of 95 °C for 15 s, 62 °C for 1 minute, and 70 °C for 1 s. Cycling was ended at 95 °C with a 5 °C/s calefactive velocity to create the melting curve. The expression level of each gene was calculated using the Livak (2$^{-\Delta\Delta CT}$) method after normalization to *GAPDH* (GenBank No. NM_001256799.3) or *EF-1a* (GenBank No. GU136229). Primers are listed in S1 Table.

## 4.6 Dual luciferase assay

HEK293T or HEK293T-MAVS-Knockout cells were cultured in a 24-well plate for dual luciferase assay. The cells of each well were transfected with 0.2 µg of Firefly luciferase reporter-gene plasmids (promoter region of IFNβ or LvVago4) [10], 0.04 µg of pRL-TK Renilla luciferase plasmids (internal control, Promega), and 0.01, 0.05 or 0.1 µg of protein expression plasmids or 0.5 µg of pcDNA-HA plasmids (as a control). Then, 48 h after transfection, the cells were treated by ISD and poly(I:C) for 6 h. The cells were lysed, and 60% of the lysis was used in measuring Firefly of Renilla luciferase activity with Dual-Glo Luciferase Assay System (Promega). The remaining 40% of the cell lysis was analyzed through Western blotting, and the expression levels of proteins were detected. All the experiments were repeated three times.

## 4.7 Pull-down assay

Pull-down assays for ectopically expressed cGAS or cGAS-like homologs binding to dsDNA or dsRNA were performed with monoclonal Dynabeads MyOne Streptavidin T1. The beads were incubated with purified protein diluent or cell lysis containing endogenous ectopically expressed cGAS or cGAS-like homologs (HEK293T cells with MAVS-knockout) which transfected with biotin-ISD (Thermo fisher scientific) or biotin-poly(I:C) (Invivogen) for 3 h at 4 ℃, and then washed with PBS five times, and finally sampled with protein loading buffer. All the samples were analyzed with SDS-PAGE for Western blotting with antibodies of anti-GAPDH (Proteintech) and anti-HA (CST).

## 4.8 Co-IP

HEK293T cells were seeded in 6-well plates at a density of $2 \times 10^5$ cells per well in complete growth medium. Transfections were performed 24 hours post-seeding using Superluminal transfection reagent (MIKX) according to the manufacturer's instructions. Specifically, a transfection mixture containing 1 µg GFP-tagged plasmid and 1 µg HA-tagged plasmid was prepared by diluting the DNA to a final concentration of 10 ng/µL in serum-containing DMEM medium. The Superluminal reagent was added at a ratio of 3 µL per µg of total plasmid DNA. The solution was vortex-mixed thoroughly and incubated at room temperature for 10 minutes to allow complex formation. The DNA-reagent complexes were then added dropwise to the cells in fresh complete medium. In brief, 48 h after transfection, HEK293T cells were harvested and washed with ice-cold PBS three times and then lysed in IP lysis buffer (Pierce) with Halt Protease Inhibitor Cocktail (Thermo fisher scientific). The supernatants (500 µL) were incubated with 30 µL of anti-GFP (MBL) at 4 °C for 4 h. The agarose affinity gels were washed with PBS five times and subjected to SDS-PAGE assay and Western blotting using rabbit anti-GFP antibody (Sigma), rabbit anti-HA antibody (Merck). Ten percent total cell lysis was examined as the input controls.

## 4.9 Extraction and purification of cellular cGAMP

Freshly prepared extraction reagent was obtained by mixing methanol:acetonitrile:ultrapure water at a volumetric ratio of 2:2:1, followed by ice bath equilibration. For internal standard-containing extraction reagent, tenofovir was dissolved in the extraction reagent to achieve a final concentration of 25 ng/mL. After removing the supernatant, 300 µL of extraction

reagent containing the internal standard was added to the culture plate. Cells were scraped and lysed, after which the lysate was transferred to a 2 mL EP tube. Two sequential 400 µL aliquots of extraction reagent (without internal standard) were used to rinse each well. Both aliquots were combined with the corresponding cell lysate in the 2 mL EP tube. Samples were heated at 95 °C for 15 min, immediately cooled on ice, and then incubated overnight at -20 °C to precipitate proteins. Protein pellets were removed by centrifugation at 20,000 × g for 15 min. The protein-free supernatant was transferred to a new EP tube and completely evaporated to dry white/red crystalline particles using a Concentrator plus (Eppendorf). The residue was reconstituted in 100 µL of HPLC-grade water and stored at -20 °C. Samples were analyzed using a triple quadrupole liquid chromatography-mass spectrometry (LC-MS/MS) system at the Sun Yat-sen University Testing Center. Retention times and polarity characteristics were compared with reference standards to confirm compound identity.

## 4.10 Detection of LvIKKε phosphorylated sites

To detect activated LvIKKε, phosphorylation-specific antibodies were developed against its phosphorylated sites. HepG2 cells were transiently transfected with pcDNA3.1-LvIKKε-GFP plasmid using Superluminal transfection reagent (MIKX) according to the manufacturer's protocol. At 24 hours post-transfection, cells were stimulated with poly(I:C) (InvivoGen; 10 µg/ml) for 24 hours to induce immune activation. Cells were subsequently lysed in IP lysis buffer (Pierce) with Halt Protease Inhibitor Cocktail (Thermo fisher scientific). Lysates were incubated with anti-GFP Magnetic Beads (MBL) for 2 hours at 4 °C to immunoprecipitate GFP-tagged LvIKKε. Beads were washed three times with ice-cold lysis buffer, and bound proteins were eluted in SDS-PAGE loading buffer. Eluted proteins were resolved via SDS-PAGE and stained with Coomassie Brilliant Blue. Target protein bands were excised, destained, and subjected to in-gel trypsin digestion (Promega). Peptides were subjected to LC-MS/MS.

## 4.11 RNA interference (RNAi)

T7 RiboMAX Express RNAi System kit (Promega) was used to generate dsRNA-LvSTING and dsRNA-GFP (for controls) with primers containing a 5' T7 RNA polymerase binding site (S1 Table). dsRNA quality was checked after annealing by gel electrophoresis. Each shrimp received an intraperitoneal injection of dsRNA at the second abdominal segment (2 µg/g shrimp in 50 µL of PBS) or equivalent volume of PBS. Hemocytes were collected from shrimp 48 h after dsRNA injection, and total RNA was extracted and assessed by qRT-PCR with the corresponding primers for the evaluation of RNAi efficacy. For survival experiment, 48 h after dsRNA injection, the shrimps were injected again with $1 \times 10^5$ copies of WSSV or DIV1 particles or mock-challenged with PBS as a control. The survival rate of each group was recorded every 4 h during 7 days after WSSV infection.

## 4.12 Immunofluorescence and confocal laser scanning microscopy

Hemocytes with different treatments, including viral infection and knockdown of specific genes, were spread onto coverslips in a 24-well plate. After 30 min, PBS was removed, and cells were fixed in 4% paraformaldehyde (diluted in PBS) at 25 °C for 15 min. The cells were then permeabilized with methanol at −20 °C for 10 min. After washing the slides three times, the hemocytes were blocked with 3% bovine serum albumin (diluted in PBS) for 1 h at 25 °C and then incubated with a mixture of primary antibodies (1:100, diluted in blocking reagent) overnight (about 8 h) at 4 °C. The primary antibodies used in immunofluorescence (IF) were rabbit anti-LvIRF antibody (Genecreate) and mouse anti-β-actin antibody (Sigma). The slides were washed with PBS six times and then incubated with diluted second antibodies of anti-rabbit IgG (H + L; 1:1000), F (ab')2 fragment (Alexa Fluor 488 Conjugate; CST), and anti-mouse IgG (H + L), F (ab')2 fragment (Alexa Fluor 594 Conjugate; CST) for 1 h at 25 °C. The cell nuclei were stained with Hoechst 33258 solution (Beyotime) for 10 min. Finally, after washing six times with PBS, the slides were observed with a confocal microscope (Leica, TCS-SP5, Germany).

### 4.13 Detection of activated LvIRF and LvIKKε

The HEK293T cell transfected the expressing plasmids or the hemocytes, gills, intestines, hepatopancreas of virus-challenged, nonchallenged, RNAi treated shrimps were collected and pooled from five shrimps for each sample at different time points, and the total proteins were extracted by RIPA lysis (Beyotime) with a protease and phosphatase inhibitor cocktail (Merck), and then centrifuged at 12,000 $g$ for 10 minutes at 4 ℃ to remove cell debris. For SDS-PAGE, protein samples were added with 5 × loading buffer and boiled for 10 min, and then subjected to Western blotting using anti-pho-LvIKKε antibodies (Genecreate) and mouse β-Actin Antibody (Abmart). For native PAGE protein samples were analyzed using 8% acrylamide gel, and further detected by Western blotting using anti-LvIRF antibodies (Genecreate).

### 4.14 Detection of viral loads by absolute quantitative PCR

Viral titers in shrimps were evaluated through absolute quantitative PCR. Briefly, we collected gills from shrimp 48 h after WSSV infection. Gill DNA was extracted with Marine Animal Tissue Genomic DNA Extraction Kit (TianGen). The quantity of WSSV genome copies was measured by absolute q-PCR using the primers of WSSV-F/R and a TaqMan fluorogenic probe named TaqMan probe-WSSV via LightCycler TaqMan Master kit (Roche) as described previously (S1 Table) [28]. The WSSV genome copies were calculated and normalized to 1 μg of shrimp tissue DNA.

### 4.15 Statistical analysis

All the data were presented as mean ± SD. Significant differences were analyzed using two-tailed Student's t-test for paired comparisons or a one-way ANOVA for multiple comparisons. Different lowercase letters indicate significant differences ($p < 0.05$) in the one-way ANOVA analysis. For survival rates, data were subjected to statistical analysis using GraphPad Prism software to generate the Kaplan ± Meier plot (log-rank $\chi^2$ test). All statistical analyses were produced using GraphPad 9.0 data view software.

## Supporting information

**S1 Fig. The examination of HEK293T-MAVS-KO.**
(TIF)

**S2 Fig. Predicted phosphorylation sites of LvIKKε.** (A) IP assay of GFP tagged LvIKKε in HepG2 cells under poly(I:C) stimulation. The protein band corresponding to the eluted products via anti-GFP Magnetic Beads was analyzed by liquid chromatography-tadem mass spectrometry (LC-MS/MS). (B) Phosphorylation probabilities of LvIKKε analyzed by LC-MS/MS. (C) Multiple sequence alignment of LvIKKε and HsIKKε.
(TIF)

**S3 Fig. LvMab21–1, LvMab21–2, and LvMab21–3 respond to poly(I:C) stimulation in hepatopancreas, gill, and intestine.**
(TIF)

**S4 Fig. LvMab21–1, LvMab21–2, and LvMab21–3 activate LvIKKε and LvIRF in hemocyte, hepatopancreas, gill, and intestine in a tissue-specific manner.**
(TIF)

**S1 Table. Primers used in this study.**
(DOCX)

**S2 Table. Proteomic identification of IP products based on LC-MS/MS analysis.**
(DOCX)

**S3 Table. Proteomic identification of candidates modified residues of LvIKKε based on LC-MS/MS analysis.**
(DOCX)

**S4 Table. Original values for all graphs.**
(XLSX)

## Author contributions

**Data curation:** Haoyang Li, Sheng Wang.

**Formal analysis:** Qinyao Li, Hao Yang, Xiaodi Wang, Airong Lv, Ranran Wang.

**Funding acquisition:** Haoyang Li, Sheng Wang, Jianguo He, Chaozheng Li.

**Investigation:** Bin Yin, Chaozheng Li.

**Methodology:** Haoyang Li, Qinyao Li, Hao Yang, Xiaodi Wang, Airong Lv, Xuanzheng Di, Ranran Wang, Sheng Wang, Bin Yin.

**Project administration:** Jianguo He, Chaozheng Li.

**Resources:** Haoyang Li.

**Software:** Xuanzheng Di, Bin Yin.

**Supervision:** Haoyang Li, Chaozheng Li.

**Validation:** Haoyang Li, Jianguo He, Chaozheng Li.

**Writing – original draft:** Haoyang Li.

**Writing – review & editing:** Chaozheng Li.

## Acknowledgments

The authors are grateful to Guangdong Hisenor Group Co., Ltd Guangzhou for providing specific pathogen free shrimp. The authors are grateful to Yaoxing Wu and Shuai Yang from Jun Cui's lab for providing MAVS-knock out 293T cell line.

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
