## [Decision Letter · Decision Letter 0]

11 Dec 2025

Crustacean Mab21 Proteins Drive Tissue-Specific Antiviral Immunity by Activating IKKε Outside the Canonical Nucleic-Acid Sensing Paradigm

PLOS Pathogens

Dear Dr. Li,

Thank you for submitting your manuscript to PLOS Pathogens. After an arduous process, your manuscript received sufficient careful consideration to provide feedback.  It has merit but does not fully meet PLOS Pathogens's publication criteria as it currently stands. Therefore, we invite you to submit a revised version of the manuscript that addresses the points raised during the review process.

We look forward to receiving your revised manuscript.

Kind regards,

Edward S. Mocarski

Academic Editor

PLOS Pathogens

Alison McBride

Section Editor

Editor-in-Chief

PLOS Pathogens

PLOS Pathogens

orcid.org/0000-0002-7699-2064

**Additional Editor Comments:**

The Li et al., manuscript investigating three shrimp cGAS homolog Mab21 proteins has been reviewed by one revieweer in addition to the Editor. For reasons that remain unclear, it has been challenging to find reviewers willing to take on this story. In the manuscript, authors provide evidence supporting an interaction directly between Mab21 proteins and a shrimp TBK1 homolog of IKKε, promoting serine 175 phosphorylation and downstream IRF–Vago4 signaling that is tissue specific to each of the family members. Unlike vertebrate and insect counterparts, these proteins do not sense dsDNA/dsRNA or produce canonical cGAMP but are shown to act as protein-based enhancers of kinase activation.

Initially, the authors identify three Mab21 protein coding sequences in the genome of white Pacific shrimp but observe that all three lack consensus canonical nucleotidyltransferase domains characteristic of cGAS-like homologs that recognize either dsDNA or dsRNA in vertebrates and other invertebrates. Authors try and fail to see any evidence that the shrimp Mab21 proteins behave as nucleic acid sensors in conventional overexpression systems, negative experiments that are included in Fig 2 and discussed later in the manuscript. Authors proceed to coimmunoprecipitation of overexpressed proteins in HEK293T cell extracts revealed that all three Mab21 proteins interact with shrimp IKKε (a homolog of TBK1) and then proceed to characterize a critical phosphorylation site (S175) enhanced by any of the Mab21 proteins again by overexpression. The artificial circumstances where this work is carried out raises concerns that are not fully addressed as the data are generated and presented. These activities should be shown to occur in shrimp or shrimp cells concurrent with overexpression work. The work in shrimp systems in necessary to consider the Mab21 observations verified.

Furthermore, given that this is about an interferon-like response, more needs to be done to show an antiviral impact akin to what is projected by authors comments. It is only in Fig. 5 where WSSV infection is shown to rapid and strong induction of LvMab21-1 ~10-fold by 36 h. This seems modest at best for interferon-type response both in induction level and kinetics and raises questions as to what signaling precedes the function of Mab21 proteins in the natural system. Nevertheless, RNAi knockdown of LvMab21-1, LvMab21-2, or LvMab21-3 is shown to significantly reduce phosphorylation of shrimp IKKε and activation of IRF and Vago4 in Fig. 6, consistent with the conclusion that “all three shrimp Mab21 proteins positively regulate the IKKε–IRF–Vago4 pathway in hemocytes”. Additional details are presented in Figs 7 and 8; however, there is too little attention to whether the pathway impedes WSSV infection dependent on the Mab21 proteins, all of which is fundamental to the authors inquiry. Besides the actual antiviral impact, the kinetics of the response should be expanded so as to understand whether this is an innate host antiviral response to virus infection that yields an active interferon.

**Journal Requirements:**

At this stage, the following Authors/Authors require contributions: Chaozheng Li. Please ensure that the full contributions of each author are acknowledged in the "Add/Edit/Remove Authors" section of our submission form.

- TM on pages: 13, 15, and 16.

Potential Copyright Issues:

i) Please confirm (a) that you are the photographer of 9C, or (b) provide written permission from the photographer to publish the photo(s) under our CC BY 4.0 license.

ii) Figures 1C, and 6A. Please confirm whether you drew the images / clip-art within the figure panels by hand. If you did not draw the images, please provide (a) a link to the source of the images or icons and their license / terms of use; or (b) written permission from the copyright holder to publish the images or icons under our CC BY 4.0 license. Alternatively, you may replace the images with open source alternatives. See these open source resources you may use to replace images / clip-art:

**Reviewers' Comments:**

Reviewer's Responses to Questions

**Part I - Summary**

Reviewer #1: In this new submission, teh authors investigate the role of cGAS like Mab21 proteins in the shrimp antiviral immune response. Surprising they fail to identify either nucleic acid binding activity nor CDN synthesizing activity. Instead they identify interactions with the shrimp Mab21 proteins (3 ISSO forms) and the IKKe homolog, leading to IKK phosphorylation/activation. They map an activating phosphorylation site in shrimp IKKe, at a location that is orhtologous to mammalian IKKe; presumably this is an autophosphoyrlation event, although that is not demonstrated.

Using RNAi approaches in shrimp, they show these Mab21 proteins are critical for IKKe phosphorylation/activation as well as IRF dimerization, with varying levels of expression/inducaton and relevance in different shrimp tissues.

**Part II – Major Issues: Key Experiments Required for Acceptance**

Reviewer #1: The authors should address, more directly, if the Mab21 have a charged surface, like cGAS and cGLRs, that bind nucleic acid. They also mention, but only in the Discussion, the shrimp Mab21s lack catalytic residues…this should all be directly demonstrated and displayed in teh Results and Figure 1.

Throughout the manuscript, when more than two groups are compared, an ANOVA type analysis should be used to analyze for signficance (t-test not sufficient).

The efffects of Mab21 silencing in hemocytes is modest (6C), at best. This and other key immunoblots should be included replicates and quantitation. The data from the hepatopancreas is much more robust. The blots from the intestines - not convincing. These blots are critical linchpins for this story and all should be more convincing and replicates shown and quantified, to demonstrate rigor and reproducibility.

The senstiivity to infection with individual RNAi targeting of each Mab21 is also modest. Teh authors should knockdown multiple (or all 3) Mab21s, as there model predicts (at least additive) effects of knocking down each, with hypothesized roles in distinct tissues.

**Part III – Minor Issues: Editorial and Data Presentation Modifications**

Reviewer #1: Suggested to reorganize data in Figure 7, so the changes with time/infection/challenge and distinction between tissues is more easily comprehended.

The first paragraph of the Discussion should be reworded to be NOT over interpret the negative conclusions regarding sensing and CDN synthesis. The following paragraph does an excellent job explaining the caveats, but the first paragraph overstates the conclusions.

PLOS authors have the option to publish the peer review history of their article (what does this mean? ). If published, this will include your full peer review and any attached files.

**Do you want your identity to be public for this peer review?** For information about this choice, including consent withdrawal, please see our Privacy Policy .

Reviewer #1: No

**Figure resubmission:**

**Reproducibility:**



---

## [Editor Report · Decision Letter 1]

9 Feb 2026

Dear Prof. Li,

We are pleased to inform you that your manuscript 'Crustacean Mab21 Proteins Drive Tissue-Specific Antiviral Immunity by Activating IKKε Outside the Canonical Nucleic-Acid Sensing Paradigm' has been provisionally accepted for publication in PLOS Pathogens.

Best regards,

Edward S. Mocarski

Academic Editor

PLOS Pathogens

Alison McBride

Section Editor

PLOS Pathogens

Sumita Bhaduri-McIntosh

Editor-in-Chief

PLOS Pathogens

orcid.org/0000-0003-2946-9497

Michael Malim

Editor-in-Chief

PLOS Pathogens

orcid.org/0000-0002-7699-2064

Thank you for revising the manuscript very carefully in response to reviewers' concerns.
---

## [Editor Report · Acceptance letter]

Dear Prof. Li,

We are delighted to inform you that your manuscript, "Crustacean Mab21 Proteins Drive Tissue-Specific Antiviral Immunity by Activating IKKε Outside the Canonical Nucleic-Acid Sensing Paradigm," has been formally accepted for publication in PLOS Pathogens.

Best regards,

Sumita Bhaduri-McIntosh

Editor-in-Chief

PLOS Pathogens

orcid.org/0000-0003-2946-9497

Michael Malim

Editor-in-Chief

PLOS Pathogens

orcid.org/0000-0002-7699-2064